# Characterization of the Erosion Damage Mechanism of Coal Gangue Slopes through Rainwater Using a 3D Discrete Element Method: A Case Study of the Guizhou Coal Gangue Slope (Southwestern China)

Yun Tian [1,2,*] , Yong Wu [1,2,*], Jiangyu Lin [1,2], Xueling Li [1,2], Dafu Xu [1,2], Futao Zhou [1,2] and Qi Feng [1,2]

1    College of Environment and Civil Engineering, Chengdu University of Technology, Chengdu 610059, China
2    State Key Laboratory of Geohazard Prevention and Geoenvironment Protection, Chengdu 610059, China
*    Correspondence: 74616.cool@163.com (Y.T.); ywu@cdut.edu.cn (Y.W.)

**Featured Application: This paper provides a simulation method for extreme precipitation events in geotechnical slopes (spatio-temporal connections, forecasting, generation, impact analysis, and vulnerability and risk assessment). Our improved methods provide a valuable tool for engineering disaster early warning and contribute to a better understanding of hydrodynamic processes in general.**

**Abstract:** Coal gangue is one of the largest solid wastes in the world. In previous studies, the influence and mechanisms of rainfall infiltration on coal gangue slope stability and possible rain erosion have been studied through theoretical analysis, numerical simulation, and modelling, and the results have indicated that discontinuous discrete element methods are the most suitable for determining the erosion mechanism of coal gangue slopes. In this study, we take a Guizhou coal gangue slope as a general case, use three-dimensional Particle Flow Code (PFC3D) as the key method, and combine discrete element fluid–structure coupling technology with optimized erosion shear failure theory to determine the erosion failure mechanism of coal gangue slopes. We investigate a coal gangue slope near the electric power plant in Panzhou City, Guizhou Province (China) as a case study, and conduct a comprehensive analysis of the erosion induced by the corrosion damage mechanism. We use the PFC3D method, combined with optimized rain erosion shear failure theory, for our investigation. The applied methods mainly consider dynamic inversion of the erosion process, as well as the changes in coordination number, porosity, unbalanced force, and energy dissipation. The scour damage type of the studied gully is intermittent fragmentary damage, with the following inferred damage sequence: Center–bottom–top of the slope. The entire erosion damage process can be divided into three stages: catchment–fracture, erosion–accumulation, and piping–penetration failure. In the first stage of erosion, the force chain fracture is the most severe. The maximum kinetic energy reaches 25 MJ and the coordination number decreases from 5.3 to 4.0, whereas the porosity increases from 0.42 to 0.45. Unexpected lateral erosion and expansion occur at 40–60 m (in the central slope) in the y-direction of the slope, the unbalanced force reaches 7500 N, and the peak porosity is increased by 10%. This paper provides a simulation method for extreme precipitation events in geotechnical slopes (contributing to spatio-temporal connections, forecasting, generation, impact analysis, and vulnerability and risk assessment). Our improved methods provide valuable tools for engineering disaster early warning, and contribute to a better understanding of hydrodynamic processes in general.

**Keywords:** particle flow; rainfall erosion; gully; coal gangue; fluid–structure interaction; hydrodynamics

## 1. Introduction

Coal gangue is a solid waste with low calorific value, being a by-product of the process of coal mining and washing. It is one of the most abundant and voluminous solid wastes discharged in the world, and its production continues to increase [1,2]. Coal gangue is commonly piled up in open-mine dumps, and not only occupies large surface areas and pollutes the environment, but also has poor engineering stability. Therefore, it is easily affected by external natural forces that may cause changes to its physical and mechanical properties. This may result in slope collapse, posing a serious threat to the safety of people and human property.

Southwest China is a high-incidence area of hydrodynamic landslides of coal gangue dumps. Damages caused by these landslides directly threaten the safety of people, property, and infrastructure, but may also cause far-reaching secondary disasters [3]. Therefore, the improvement in early warning systems and emergency response levels for disasters induced by hydrodynamic landslides of coal gangue dumps is urgently necessary. First, the processes related to rain erosion damage must be solved, as rain erosion is the primary major factor in the damage of coal gangue slopes. Seepage and water erosion gradually induce deformation and failure of the slope. In this way, the stability continuously decreases, tends towards the limit instability state, and finally leads to an overall instability failure under changing short-term hydrological conditions. The stability analysis of high coal gangue slopes has become an important research topic in geotechnical engineering in recent years. However, to date, the most important factor in slope instability—the erosion failure mechanism—has only been poorly studied. Therefore, our purpose is to determine the inherent mechanism of slope disasters under extreme rainfall conditions, in order to enhance the effect of prediction and management.

At present, many scholars tend to use continuum methods (e.g., FEM, FDM, and PFEM) in their research, but these methods present certain drawbacks; for example, they often rely on highly simplified constitutive equations that specify properties. However, there are many parameters in the constitutive equations for accurate prediction, and it is very difficult and cumbersome to calibrate these parameters. In the process of calculation, it is often difficult to calculate convergence when considering plastic problems. On the other hand, the continuum calculation method cannot effectively handle, calculate, and simulate fracturing and large deformation problems. In addition, in recent years, the SPH (Smoothed Particle Hydrodynamics) method has become popular, both at home and abroad, in the study of fluid motion. This method is a meshless adaptive Lagrangian particle method [4]. It is more complicated, and is not suitable for the simulation of monitoring points for slope stress and strain. It should be noted that the smooth search function plays an important role in the SPH method, as it determines the precision and computational efficiency of the functional expression; however, such a function requires a complex algorithm and time to complete, and there are certain additional drawbacks [5].

The advantage of PFC is that it can simulate material transport and stress transfer in soil, as well as the deformation, expansion, and extension of rocks and soil in a relatively simple way. It can monitor the position and number of particles generated during the failure of a sample, the stress and strain inside the sample, the shape of the soil when it is deformed, and the strength of the model. The process of dynamic destruction of the macroscopic model can also be observed. Accordingly, this method can realize observations at both the micro- and macro-scale for soil simulation material mechanical and model testing.

The influence and mechanism of rainfall infiltration on the slope stability and possible rain erosion have been studied through theoretical analysis, numerical simulation, and modelling in previous studies. These studies have mainly focused on the influence of rainfall infiltration on the slope stability [6–9], slope erosion characteristics [10,11], and gully erosion generation mechanism [12,13], as well as factors influencing erosion, such as rainfall [14–17], slope length [18], and slope angle [19]. Enrico Conte et al. [20] have obtained material parameters based on some closed-form expressions derived using physics-based models, and considered two typical triggering mechanisms to predict shallow landslides

under expected rainfall scenarios. E. Napolitano et al. [21] have identified a method to explain the different hazards associated with seasonality of hydrological processes within ash-fall pyroclastic mantle based on landslide physical model reconstruction, in situ hydrological monitoring, and hydrology and slope-stability-modeling conditions. Silvia Peruccacci et al. [22] have reconstructed the rainfall history that could have contributed to slope damage, determining the corresponding rainfall duration D (in hours) and cumulative event rainfall E (in mm). Then, using a power-law threshold model, the cumulative event rainfall–rainfall duration (ED) threshold was determined. Combined with FLAC2D, Pasculli Antonio et al. [23] have combined Monte Carlo technology with an FDM (finite difference method) to study the stability of an actual slope, which provided reference for more accurate modeling and monitoring of rock and soil masses. Among the proposed numerical methods, the discrete element method combines the fluid module with the soil from the perspective of micro-dynamics, considers the interactions between water and soil, and can dynamically evolve the large deformation characteristics of the soil during rainfall. Therefore, discontinuous discrete element can be considered the most suitable method for deciphering the erosion mechanism of a coal gangue slope. To date, the specific formation mechanism of the erosion failure of coal gangue slope has not been studied using a three-dimensional discrete element method. Zuo et al. [24] have used three-dimensional particle flow (PFC3D) discrete element numerical software to determine that the hydrostatic pressure is closely related to the stability of the slope, and obtained threshold rainfall warning parameters. Ke et al. [25] have used PFC2D to simulate the scouring and spalling process of a steep slope. Wu et al. [26] have applied the PFC3D software to simulate the process of rainfall scouring on loess slopes with fluid–solid coupling, and compared the results with the experimental results from a laboratory test. Zhang [27] has conducted fluid–solid coupling simulation using PFC2D. The slope rate suitable for the stability of the roadbed against rain erosion was detected through simulation. However, the erosion mechanism of coal gangue slopes remains poorly understood. A review of the previous literature indicated that discrete element technology has not been applied to decipher the formation and evolution process of the main gullies. Moreover, the discrete element fluid–structure coupling technology has not yet been combined with shear erosion theory. In the present study, therefore, we use PFC3D as the key method, combined with optimized erosion shear failure theory, in order to determine the erosion failure mechanism of coal gangue slopes.

The key contributions of this paper are as follows: 1. The internal mechanism by which rainfall damages coal gangue slopes is derived through the simulation of dynamic damage, where the method of energy is introduced to conduct in-depth research on the damage mechanism; 2. we optimize the scouring shear failure theory and combine it with the discrete element PFC3D method, thus verifying the rationality and providing a new idea for PFC fluid–structure interaction methods; and 3. we enrich the PFC modeling methods for coal gangue slope modeling, providing valuable reference for the early warning and simulation of coal gangue slope engineering disasters.

## 2. Erosion Theory

### 2.1. Description of the Study Area

We selected a coal gangue slope near the Panzhou Electric Power Plant in Guizhou Province (China) as a case study for our investigation. According to the field survey, the upper lithological layer in the study area is the Quaternary Holocene artificial accumulation layer (Q4ml) with loose black coal gangue. The lower layer is gray-white limestone of the Lower Triassic Yongningzhen Formation (T1yn). The coal gangue deposited in the tailings dam reservoir area is mainly located on the left side slope of the dam area, constituting a total volume of about 670,000 m$^3$ (Figure 1). The elevation at the top of the slope is 1710.0 m, its height is about 40–60 m, and the slope is inclined by 25–40°. The coal gangue slope is unstable, representing a serious safety hazard to the ash yard and the residential buildings downhill of the mountain. The lithology of the slope is dominated by an artificial

accumulation of coal gangue with a very loose structure, especially at the corner of the slope (the yellow oval area in Figure 1). During erosion through rainwater, the water accumulates in a natural gully at the corner along both sides of the slopes, forming the largest gully (termed No. 3 gully), with a width of 3–10 m, a length of about 90 m, and a maximum depth of 8 m (Figure 2). Therefore, we focused our investigation on the vicinity of the No. 3 gully.

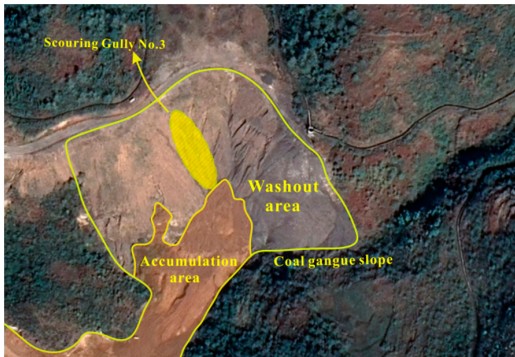

**Figure 1.** Satellite map of the studied coal gangue area in the Guizhou Province (China).

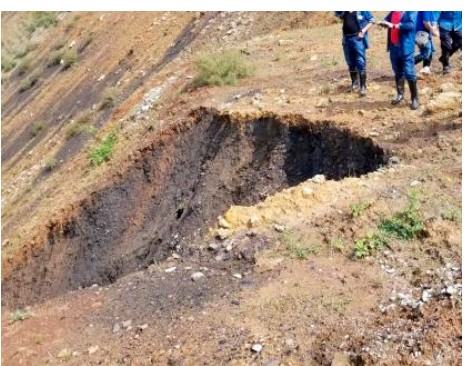

**Figure 2.** Status of No. 3 gully.

*2.2. Shear Erosion Principle*

We subdivided the coal gangue slope near the No. 3 gully and established erosion shear and rainwater infiltration models (Figures 3 and 4). The slope model had a height of about 66 m, a width of 66 m, and the incline of the slope was 25–38°. As erosion mainly occurs during heavy rainstorms, the erosion parts are mainly concentrated in the natural gully (near the No. 3 gully). We ignored the raindrop splash erosion in the early stage of rainfall, and only considered the water flow erosion and infiltration of the No. 3 gully on the slope surface.

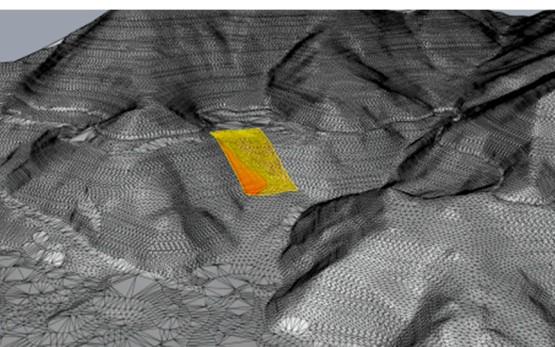

**Figure 3.** Three-dimensional model of the slope terrain.

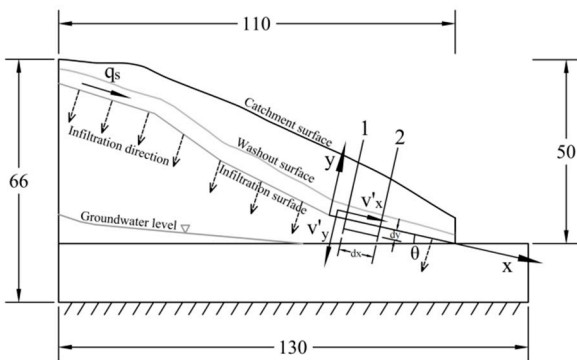

**Figure 4.** Shear erosion theory model.

We used the study by Shen et al. [28], on the erosion shear theory of gravel soil, in order to calculate the erosion depth of the soil over time for derivation of the erosion theory. However, the considered speed factor includes a certain error range, which cannot match the particle mesoscopic motion speed in discrete element PFC and, therefore, needed to be refined. During heavy rain, the erosion phenomenon of the viscous gravel soil slope presents a turbulent water flow pattern. The water flow is affected by the soil particles on the slope, forming many vortices of different sizes, with positions, shapes, and flow rates constantly changing as it progresses. At first, we express the shear stress of the slope surface flow [29] using the following equations:

$$\tau_w = \tau_{wa} + \tau_{wb}, \tag{1}$$

$$\tau_{wa} = -\rho v'_x v'_y, \tag{2}$$

where $\tau_{wa}$ is the viscous shear stress in laminar motion, $\tau_{wb}$ is the turbulent shear stress, $\rho$ is the flow density, and $v'_x$ and $v'_y$ as the pulsating flow velocities in the $x$ and $y$ coordinate directions, respectively (Figure 4). Defined by the pulsation speed, the velocities are calculated from the following equations:

$$\begin{cases} v'_x = v_{x0} - \overline{v}_x \\ v'_y = v_{y0} - \overline{v}_y \end{cases}, \tag{3}$$

where $v_{x0}$ and $v_{y0}$ are the instantaneous velocities of soil particles, and $\overline{v}_x$ and $\overline{v}_y$ are the average water flow velocities. According to the definitions of water flow velocity and infiltration velocity, the pulsating infiltration velocity in the y-direction is expressed by the equation:

$$v'_y = v'_x \cdot \xi = (v_{x0} - \overline{v}_x) \cdot \xi, \tag{4}$$

where $\xi$ is the effective porosity between particles. According to the assumption of Prandtl and the hydraulic roughness test of Nikuradse, the following equation can be derived [30]:

$$\tau_{wb} = \left( \frac{\overline{v}_x}{2.5ln\frac{R}{\mu} + 4.75} \right)^2 \rho, \tag{5}$$

where $R$ is the hydraulic radius and $\mu$ is the slope roughness. According to previous experimental studies [31,32], the surface roughness can be described by the average particle size of soil particles (i.e., $\mu = \overline{d}$, where $\overline{d}$ is the average particle size of PFC particles). Substituting Equations (2)–(5) into Equation (1) yields:

$$\tau_w = \left( \frac{\overline{v}_x}{2.5ln\frac{R}{\overline{d}} + 4.75} \right)^2 \rho - (v_{x0} - \overline{v}_x)^2 \cdot \xi \rho = \rho \left[ \left( \frac{\overline{v}_x}{2.5ln\frac{R}{\overline{d}} + 4.75} \right)^2 - \xi(v_{x0} - \overline{v}_x)^2 \right]. \tag{6}$$

The instantaneous velocity of soil particles can be monitored in the discrete element PFC calculation, while the average velocity of the fluid and the effective porosity of the particles are known quantities. Therefore, the optimized shear stress of water flow (Equation (6)) can be applied to the simulation calculation in PFC. For rock and soil slopes, the erosion ability of the slope surface should consider the erosion intensity of water flow. Coal gangue is a relatively coarse scattered rock and soil particle, which generally appears as agglomerates, lumps, or flakes during the erosion process. Therefore, the scouring rate $g_p$ of the coal gangue slope can be expressed as:

$$g_p = \frac{0.015 \rho_c g q_s J^{0.5} \left( \frac{\tau_w + \tau_c}{\tau_w} \right)}{tan\varphi cos\theta}, \tag{7}$$

$$q_s = \frac{h_c}{n} R^{\frac{2}{3}} J^{\frac{1}{2}}, \tag{8}$$

$$\tau_c = \gamma h_c J, \tag{9}$$

where $\rho_c$ is the soil particle density on the slope surface, $q_s$ is the single-width flow rate, the Manning coefficient $n = 0.025$, the hydraulic radius $R = h_c$, $J = sin45° = 0.706$, $\tau_c$ is the critical drag force, $\gamma$ is the water weight (with a value of $1 \times 10^4$ N/m$^3$), $h_c$ is the slope run-off depth, $\varphi$ is the internal friction angle of the soil, and $\theta$ is the slope angle.

According to the continuity equation of soil particles on the slope, it follows that:

$$\frac{\partial g_p}{\partial x} + \rho'_c g \frac{\partial y}{\partial t} = 0, \tag{10}$$

where $\rho'_c$ is the dry density of soil particles, defined by the equation:

$$\rho'_c = \rho_c (1 - \xi). \tag{11}$$

Changing Equation (10) into another expression and expressing it as a differential expression, we obtain:

$$\frac{\Delta g_p}{\Delta x} + \rho'_c g \frac{\Delta y}{\Delta t} = 0, \tag{12}$$

$$\Delta y = -\frac{\Delta g_p \Delta t}{\rho'_c g \Delta x} = \frac{(g_{p2} - g_{p1}) \Delta t}{\rho'_c g \Delta x}. \tag{13}$$

Substituting Equations (7)–(9) into Equation (13) yields:

$$\Delta y = \omega \left( \frac{1}{\tau_{w1}} - \frac{1}{\tau_{w2}} \right) \frac{\Delta t}{\Delta x}, \tag{14}$$

$$\omega = \frac{0.015 q_s J^{0.5} \tau_c}{0.6 tan\varphi cos\theta}, \tag{15}$$

where $\omega$ is the scouring coefficient (which is related to the water depth of the slope, the slope angle of the slope, and the friction angle in the soil), $\tau_{w1}$ and $\tau_{w2}$ are the water flow shear stresses of Sections 1 and 2, respectively, and $\Delta x$ is the distance between Sections 1 and 2 on the slope surface. Equation (14) can be reduced to the equation:

$$\Delta y = k \Delta t, \tag{16}$$

$$k = \omega \left( \frac{1}{\tau_{w1}} - \frac{1}{\tau_{w2}} \right) \frac{1}{\Delta x}, \tag{17}$$

where $k$ is the scouring intensity, which represents the thickness of the soil layer eroded by water flow in a time unit and is related to the position of the slope.

Finally, according to the above equations, the optimized erosion depth expression at any position of the soil over a time period can be obtained using the equation:

$$\begin{cases} \Delta y = \frac{0.015 q_s J^{0.5} \tau_c \Delta t}{0.6 tan\varphi cos\theta} \left( \frac{1}{\tau_{w1}} - \frac{1}{\tau_{w2}} \right) \frac{1}{\Delta x} \\ \quad = \frac{0.5 \gamma h_c^{\frac{8}{3}} (\tau_{w2} - \tau_{w1}) \Delta t}{tan\varphi cos\theta \tau_{w1} \tau_{w2} \Delta x} \\ \tau_{w(1,2)} = \rho \left[ \left( \frac{\overline{v}_x}{2.5 ln \frac{K}{d} + 4.75} \right)^2 - \xi (v_{x(1,2)} - \overline{v}_x)^2 \right] \end{cases}. \tag{18}$$

### 2.3. Construction of the Erosion Model in PFC 3D

The PFC particle flow method regards materials (e.g., rock, sand, and clay) as discontinuous, and regards the particulate matter composing the material as an independent basic unit. The interactions between particles reflect the macroscopic mechanical properties of the material. The particle flow numerical analysis method defines a material as a particle aggregate composed of finite particles, where a particle is a rigid body with mass; in three dimensions, a particle is a spherical particle with a unit mass. The basic idea of the discrete element method is to obtain the generalized unbalanced force of the particle through the force–displacement law, and then use Newton's second law to calculate the particle motion behavior under the action of the unbalanced force. In particular, the particle flow method uses Newton's second law alternately with the force–displacement law to perform cyclic calculations.

In the discrete element simulation research, because the size of the actual soil particles is too small, it is almost difficult to use the actual soil particles for numerical calculation. Therefore, in the PFC simulation, the particle size is often enlarged, and the specific magnification factor is usually determined based on the needs of the research itself and the computing resources of the hardware. The dimensions of the numerical model were 130 m × 66 m × 66 m (length × width × height), considering the actual size. The particle size of the slope was correspondingly increased to avoid the problem of non-convergence and underestimation of the efficiency caused by the huge number of particles and the disparity of the particle size. Studies on the effect of the particle size [33–39] have shown that, for ratios between the model size and an average particle size > 30, appropriate changes in particle size will not affect the results of the simulation. The designed model was repeatedly checked and debugged. On the basis of improving the calculation efficiency and ensuring the rationality of the rainfall damage model, the average particle size of the particles was 1.9 m, the ratio of the minimum model size (66 m to 1.9 m) was 35, and the maximum model size was 35. The ratio of size (130 m to 1.9 m) was 68. Both data were much larger than the standard value of 30, so the particle size in this model was considered to be reasonable. At the same time, these ratio data are determined without affecting the simulation accuracy and taking into account the speed of numerical calculation. The particle size of the coal gangue soil triaxial test used in the meso-parameter calibration in this paper is consistent with the particle size of the final coal gangue slope numerical model. The total number of particles in the model was 12,786. The particles with different gradations in the model are shown with different colors in Figure 5, in order to visualize the gradation of different particles. Combined with an indoor gradation test of soil, the particle gradation in PFC is shown in Table 1. The meso-parameters of the model are summarized in Table 2, according to the debugging of the triaxial test (Figure 6).

**Table 1.** Particle gradation table.

| Granules | Coarse Gravel | Medium Gravel | Fine Gravel |
|---|---|---|---|
| Particle size (m) | 2.2–2.4 | 1.4–1.8 | 1.2–1.4 |
| Amount (%) | 15.22 | 69.43 | 15.35 |

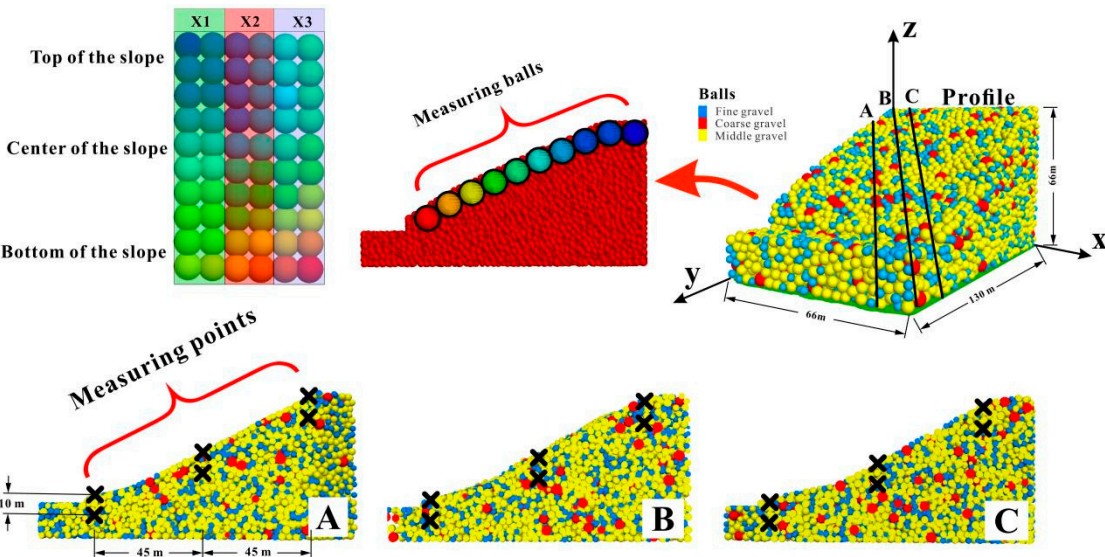

**Figure 5.** PFC 3D model and monitoring scheme.

**Table 2.** Particle Flow Model Meso-parameters.

| d/kg·m$^{-3}$ | E*/GPa | K* | μ | λ | K$_c$* | E$_c$*/GPa | $\overline{\sigma}_c$/MPa | $\overline{c}$/MPa | $\overline{\varphi}$/° | d$_w$/kg·m$^{-3}$ | v$_s$ |
|---|---|---|---|---|---|---|---|---|---|---|---|
| 3000 | 1.3 | 1.0 | 0.8 | 1.01 | 1.0 | 10 | 9 | 12 | 28 | 1.0 | $2.8 \times 10^{-2}$ |

$d_w$, fluid density; $v_s$, fluid viscosity coefficient.

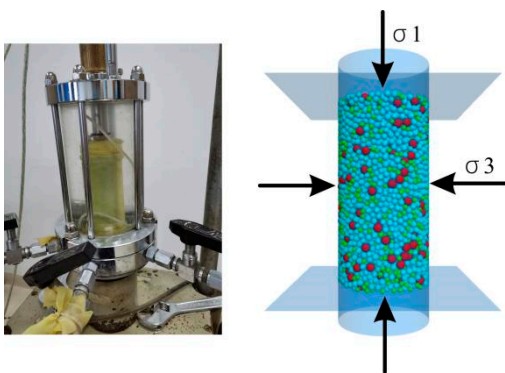

**Figure 6.** Triaxial test and meso-parameter debugging.

The data monitoring and output in the present study were mainly realized through measuring balls and monitoring points (Figure 5). We set up 60 measuring balls with a radius of 5.5 m, which were evenly distributed on the entire slope surface to monitor the real-time changes in soil porosity and coordination number over the entire slope surface, following the method described by Tian [40]. The area comprising the measurement balls was subdivided into three areas (X1, X2, and X3) in the direction of the x-axis. In the y-axis direction, it was divided into top, middle, and bottom of the slope. We also divided the No. 3 gully area into three sections, corresponding to the numbers A, B, and C. Six monitoring points were positioned at the top, middle, and bottom of the slope. Among them, two points on the same vertical line were 10 m apart. This set-up can accurately monitor the changes in the force and motion trajectory on both sides of the gully and the position of the central axis.

We simulated the flow field by establishing a fluid module (Figure 7) to set up the fluid–structure interaction scheme. To pursue a more realistic erosion simulation effect, we divided the flow field into two layers, based on the settings for the flow field constrained in previous studies [26]: the erosion layer and the permeable layer. The thickness of

the two layers was set to 0.5 m, and it continued downward with the failure depth of soil particles until the expected depth and a new state of soil equilibrium was attained. The permeable layer will weaken the bonding strength of the particles and accelerate the formation of the gully. It was assumed that the flow rate of the permeable layer is zero. The erosion layer was further divided into catchment and erosion areas. The main flow velocity was concentrated in the light green erosion area in Figure 6, as the flow field was single and could not perfectly restore the complex turbulent process in the flow field when PFC simulates large-scale fluid–structure interactions. Therefore, the particles in the non-study area were scattered, thus affecting the accuracy of the final data. The final destruction mode was disordered and difficult to observe. As the No. 3 gully is located on the trough line of the entire slope, it is a natural gully with strong water catchment capacity. Therefore, the damage depth of erosion was significantly larger than that for the catchment areas on both sides of the gully. Consequently, we ignored the impact of erosion damage caused by the adjoining catchment areas and reduced the strength of fluid mesoscopic parameters in the catchment area. A certain deviation was expected to occur between the flow velocity in the PFC and the actual slope runoff velocity. If the flow velocity is too high, the roughness of the slope will cause the collision of particles among each other, which subsequently bounce and fly off the slope. If the speed is too small, the desired effect will not be achieved, due to cohesion between the particles. A scouring speed of 2.6 m/s was determined through trial and error. After erosion of the particles on the slope, the original equilibrium was broken and the particles began to slide off the slope gradually with the fluid movement, accumulating at the bottom of the slope until a new equilibrium state was attained. At this point, the particles were immobile and the destruction process was terminated. Based on several trial calculations, we detected that the particles attained static conditions after 60 min with a maximum speed of less than $10^{-3}$ m/s.

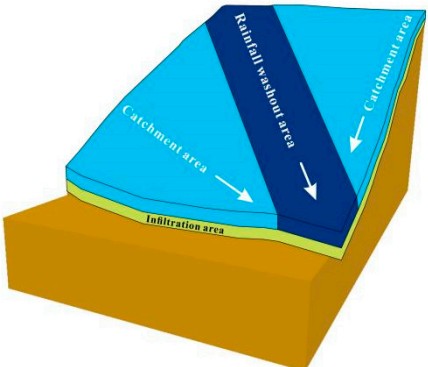

**Figure 7.** Fluid–structure interaction scheme.

## 3. Destruction Mode of the Scour Model

The failure process of the PFC fluid–structure interaction erosion model within 60 min is shown in Figure 8. The figure documents the following failure sequence of the gully: middle–bottom–top of the slope. The final damage pattern is basically the same as that for the actual No. 3 gully (Figure 8). After 10 min of erosion, the surface particles showed a striking displacement trend (Figure 8). Major displacement was concentrated in the valley at the top of the slope, which gradually expanded downward, with a maximum displacement of 0.53 m. However, the particles on the slope did not break off the slope, indicating that the soil on the slope was still in the early stage of water catchment, and the erosion effect was mainly concentrated at the top of the slope. At this stage, a runoff channel gradually formed (Figure 8). As the particles were continuously affected by the water flow, a valley runoff was basically formed after 20 min of erosion. A force chain network of the model is illustrated in Figure 9. Figure 8 shows the catchment effect on both sides of the catchment area, and the top gradually dissipated and progressed towards the center of the slope. The maximum displacement of particles reached 1.61 m at this position,

and a small number of particles started to disrupt the force chain and move down the slope with the flow of water. The contact model changed from the initial contact bond model with cohesion to the anti-rotation linear model without cohesion, indicating that the slope and valley runoff at this time had a specific soil-carrying capacity. With the gradual erosion of the particles in the center of the slope, the collision energy of the particles caused by the erosion gradually increased after 30 min, concentrated towards the bottom of the slope. At this time, the maximum displacement of the particles abruptly increased to 47.37 m, as an anti-rotation linear model had formed at the bottom and top of the slope. The particles were destroyed through collision and rubbing with the particles accumulated in the center of the slope. Moreover, the central part of the slope tended to expand in the x-direction outside the main gully. After the fluid scoured for 40 min, a maximum displacement of 96.5 m had been obtained (Figure 8). Intense denudation started to affect the top of the slope, and the particles accumulated at the bottom of the slope. At this time, the erosion pits and cavities of the slope, as well as the valley runoff, had basically formed. The anti-rotation linear model mainly focused on the top and bottom of the slope. In addition, small pipes were formed at the top of the slope, where the soil particles are in an unstable state. After 50 min of the simulation, the gradual erosion and destruction of the particles at the top of the slope caused washing of the fine particles to the bottom of the slope, where they successively accumulated. Due to cohesion, the remaining particles were not destroyed but, instead, led to a piping phenomenon. This phenomenon accelerated the destruction of the top of the gully, causing a concentration of destruction energy at the top of the slope. In addition, lateral erosion occurred in the central section of the slope, indicating an increased concentration of the energy of erosion in this region. Upon termination of the calculation after 60 min, the remaining particles at the top of the slope had completely broken and a continuous gully was formed, with a peak displacement of 140 m (Figure 8). The force chain network documented that the distribution of many anti-rotation linear models was discontinuous (Figure 9). The movement trajectory of the particles was also intermittently damaged. Our data indicate that the erosion damage type of the No. 3 gully was intermittent fragmentary damage. The following failure sequence was obtained: initial fracture in the center of the slope, the accumulation of continuous fractures at the bottom of the slope and, finally, piping and failure at the top of the slope, successively forming a large-scale valley-type gully.

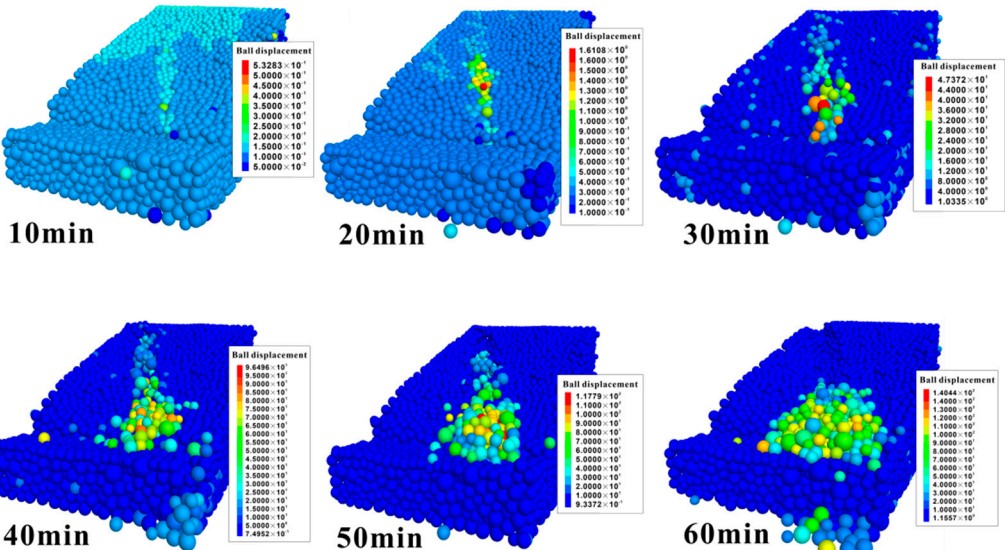

**Figure 8.** PFC3D model destruction process.

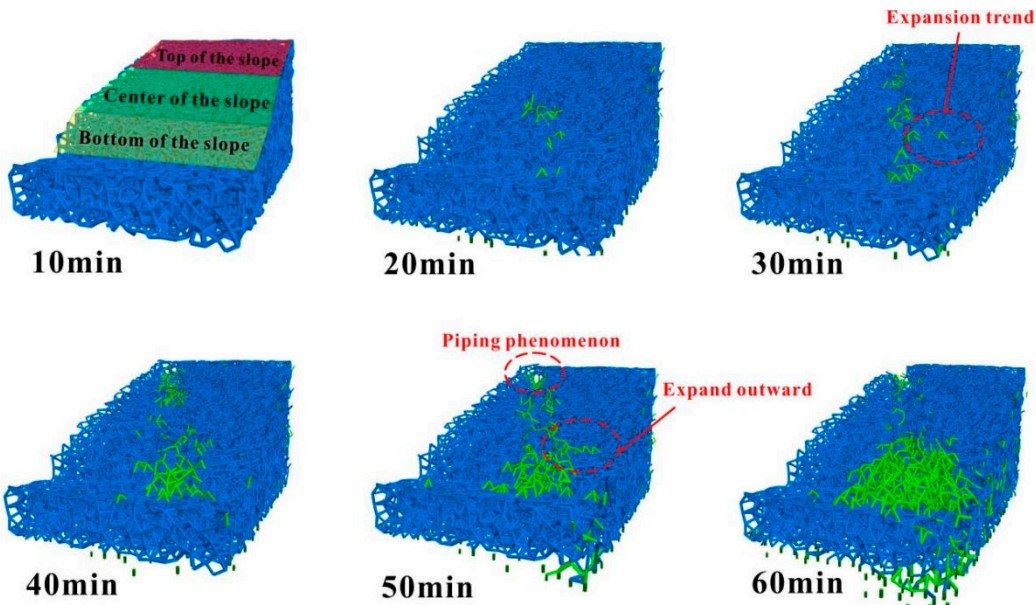

**Figure 9.** PFC3D force chain mesh destruction.

Figure 10 shows that the slope–valley gully simulated by PFC was basically consistent with the No. 3 gully in the field, and that the accumulation areas and erosion pits were in reasonable agreement. The final failure map of Section B is shown in Figure 11. To further verify the applicability of the theoretical formula and the numerical simulation of fluid–solid coupling, we considered the final failure form in the y direction as a standard and compared the erosion depth of the erosion pit, as calculated by the two methods, with the erosion depth measured in the field (Figure 12). Figures 11 and 12 indicate that the most serious erosion damage occurred between y-direction distances of 44 and 62 m from the mid-slope position, with depths of 10.2 m and 9.8 m, respectively. The second-most serious damage was located at the top of the slope, where the scouring depth reached 9.9 m. Due to the accumulation of particles at the bottom of the slope, the degree of erosion was minimal. The depth calculated by the optimized erosion theory was in good agreement with the depth of the gully obtained from the PFC simulation, proving the applicability of the two methods. Moreover, the results of the two calculation methods were consistent with the actual on-site damage situation (Figure 12), demonstrating that the proposed modelling methods represent a valuable new tool for further engineering disaster early warning and a better understanding of hydrodynamic processes in general.

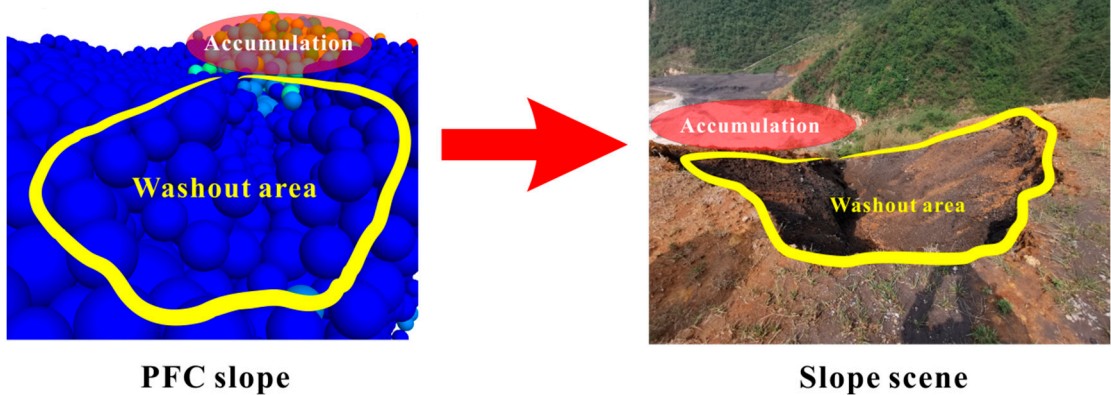

**Figure 10.** Comparison between modelled No. 3 gully and its field appearance.

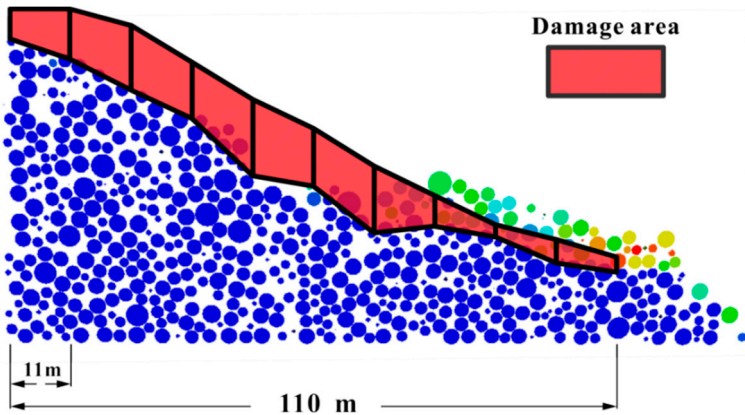

**Figure 11.** Schematic diagram of the failure depth of section B.

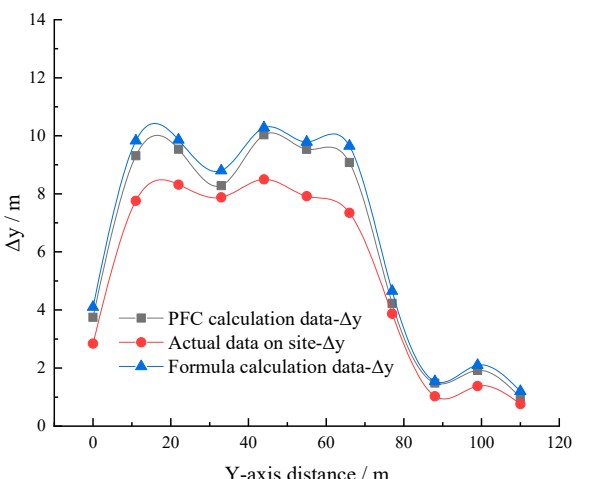

**Figure 12.** Calculated, simulation, and real data depth comparison curves.

## 4. Numerical Analysis

To further analyze the microscopic failure mode of the gullies, the changes in coordination number and porosity monitored by numerical simulations are discussed in the following. The coordination number is the average contact number of particles, which equals twice the number of particle contacts divided by the number of particles. This parameter reflects the physical properties of the particle contact model, and is closely related to the contact strength [41,42]. In the PFC3D erosion failure simulation, the coordination number can reflect the breaking time and strength decay of the specific particle contact force chain. At first, we considered the coordination number and porosity data of measuring balls Nos. 16, 22, and 59 to represent the data at the bottom, center, and top of the slope, respectively. The resultant curves are shown in Figures 13 and 14. The figures indicate that the particles in the center of the slope preferentially collided violently and were ultimately destroyed at the beginning of erosion. The coordination number decreased from 5.3 to 4.0, while the porosity increased from 0.42 to 0.45. During the following 20 min, the coordination number and porosity at this position fluctuated largely. The coordination number ranged from 4.0 to 4.65 and the porosity from 0.424 to 0.47 (Figures 13 and 14). The coordination number and porosity at the bottom and top of the slope remained relatively stable during this period, with the porosity at the slope top slowly increasing by 0.02 in the first 3 min. Subsequently, the top of the slope was slowly destroyed in the period from 20 to 37 min. The coordination number decreased from a maximum of 5.65 to 4.57, whereas the porosity concomitantly increased slowly. In contrast, the coordination number at the foot of the slope remained almost constant, as no particle damage occurred. The

central part of the slope was still in the stage of continuous erosion during this time, and the coordination number decreased rapidly by 0.25. However, the porosity in the slope concomitantly increased slowly, due to accumulation at the top of the slope. The change in porosity does not fully represent the process of fracture decay in the bond model. The force chain between the bottom and the top of the slope was obviously disrupted during the final 37–60 min, and the coordination number decreased by 1.0–1.2. The coordination number at the top of the slope began to decay rapidly at 37 min, and decayed by 0.75 within the following 8 min. Subsequently, it began to fluctuate between 0.22 and 0.25. The failure time at the bottom of the slope was about 10 min delayed, compared to that of the top of the slope, and its coordination number decayed by 1.2 between 47 and 55 min. The porosity at the top of the slope increased sharply, due to the failure of the penetration of the No. 3 gully, and increased by 0.38 within 14 min. The porosity at the top of the slope changed most significantly in the final stage of erosion, but the attenuation of the force chain was lower than that at the bottom of the slope, due to the accumulation of particles at the bottom of the slope.

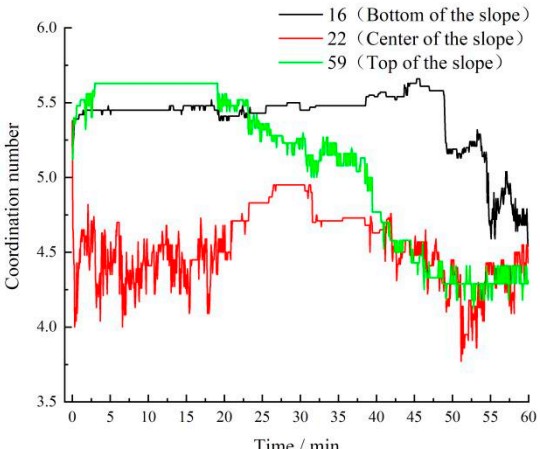

**Figure 13.** Variation in coordination number in No. 3 gully with time.

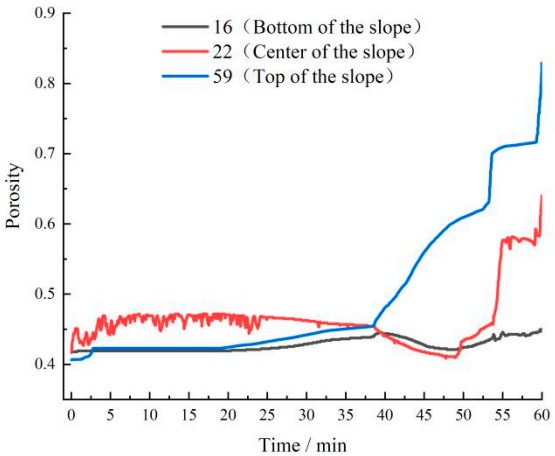

**Figure 14.** Variation in porosity in No. 3 gully with time.

As many measuring balls were used and the calculated data changed at any given time, peak porosity analysis was conducted to visualize the overall porosity change of the slope (Figure 15). The peak porosity gradually increased at the top of the slope in the X1 measurement area, reaching a maximum of 83%, gradually decreasing to 45% along the slope. The peak porosity of the X1 area changed the most dramatically, due to the larger slope than in other areas. The piping phenomenon at the top of the slope was in sharp

contrast with the particle accumulation phenomenon at the bottom of the slope, causing a large difference in porosity at both ends. As the X2 slope was in the most severely damaged area of the entire slope, the peak porosity in the center of the slope increased in the X2 area by up to 20%, forming an "M"-shaped curve distribution in the y-direction. A lateral expansion failure at 40–60 m (mid-slope) in the y-direction occurred in the X3 measurement area, with a 10% increase in peak porosity.

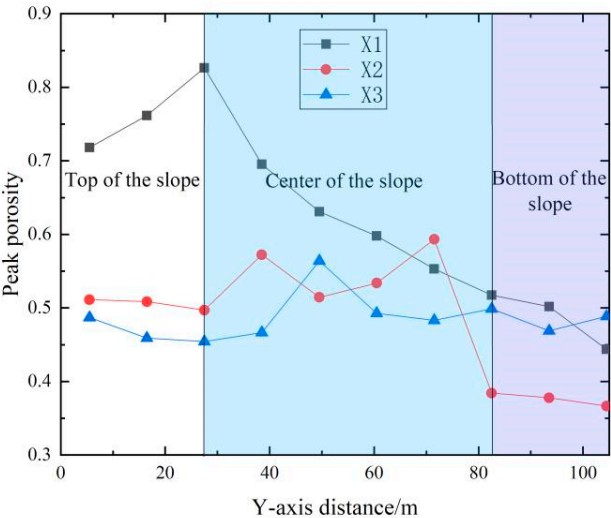

**Figure 15.** Overall slope peak porosity.

To study the interaction and expansion of grains on both sides of the gully, we selected Sections A and C as case studies. Through the unbalanced force in the x-direction measured at the monitoring point, we obtained the force curve, as shown in Figure 16. From the top to the bottom of the slope, an increasing number of particles were accumulated, due to the rising potential energy of erosion; the value of the unbalanced force gradually increased, and the degree of fluctuation gradually grew. The unbalanced force at Section A increased from 0–6000 N to 0–9600 N (Figure 16a–c), and the unbalanced force at Section C increased from 0–6100 N to 0–16,000 N (Figure 16d–i) from the top to the bottom of the slope. Therefore, the unbalanced force, which acts laterally in the model, depends on the erosion potential energy and the position of the slope. The potential energy and the effect of the force continuously rise with progression toward the bottom of the slope. The unbalanced force of profile A (representing a steeper slope) was generally lower than that of profile C, as the water flow collected at profile A will eventually act on profile C. Furthermore, the action direction of the water flow was more inclined in the x-direction. Implementing the findings from the previous section, lateral erosion and expansion in the positive x-direction will occur near point C2 in the center of the slope. Figure 16e shows that the unbalanced force reached 7500 N, which may cause lateral tributaries to expand in the gully if the water flow is eroded for about 37 min. However, the maximum unbalanced force at the bottom of the slope reached 16,000 N, without any preferential lateral expansion, as the accumulation at the bottom of the slope aggravates the collision of particles. Moreover, the direction of water flow becomes more disordered and the stress is undirected. The occurrence of lateral expansion is also related to the direction of water flow of the catchment, which is further analyzed in a later stage.

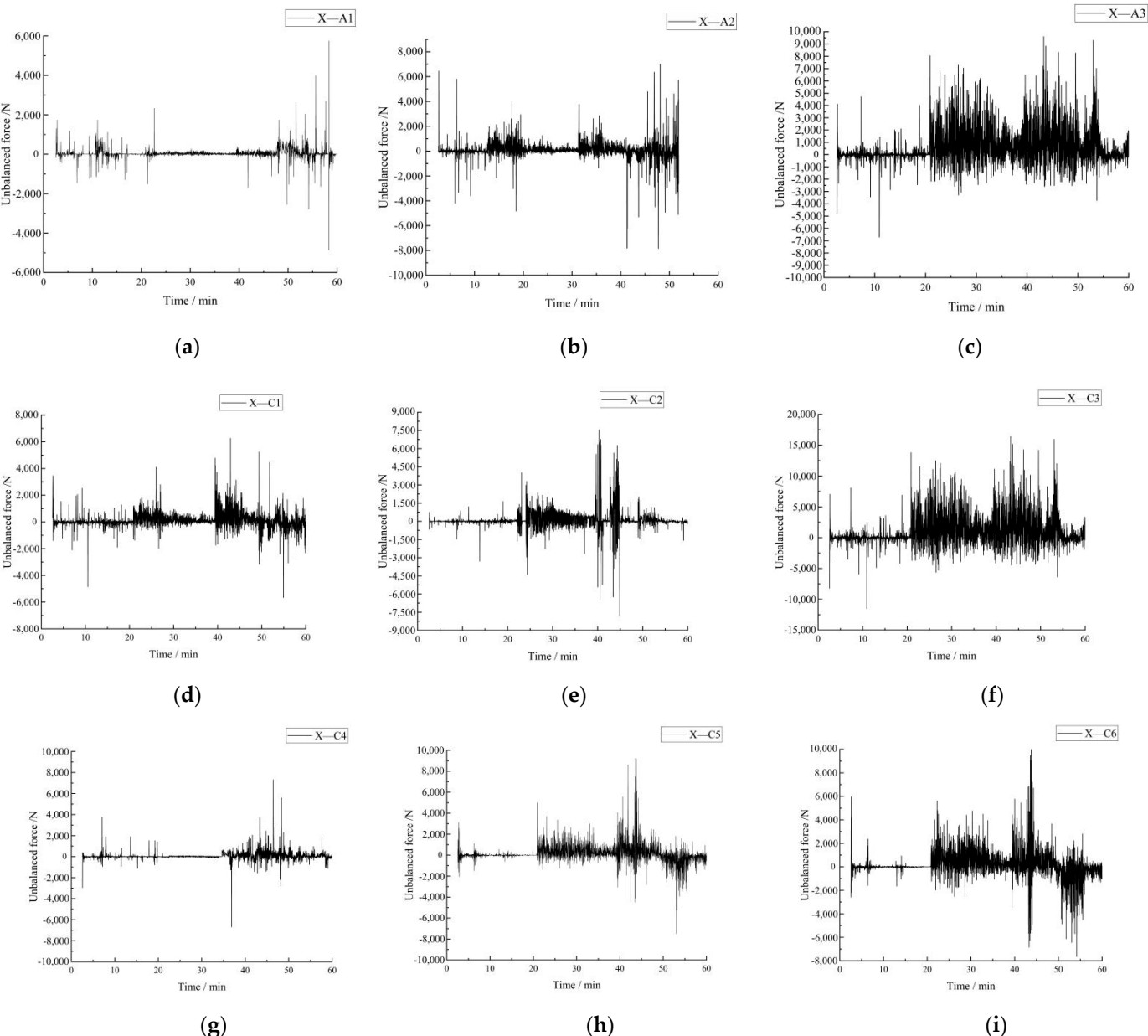

**Figure 16.** Unbalanced force curves. (**a**) Monitoring point A1 (**b**) Monitoring point A2 (**c**) Monitoring point A3 (**d**) Monitoring point C1 (**e**) Monitoring point C2 (**f**) Monitoring point C3 (**g**) Monitoring point C4 (**h**) Monitoring point C5 (**i**) Monitoring point C6.

Figure 16d–i show that the fluctuation frequency of the unbalanced force in the surface layer of the soil was larger than in the deep layer. However, the unbalanced force in the deep layer was about 800–1100 N larger than that in the surface layer on the same vertical plane, indicating that the surface particles were more susceptible to water erosion. However, this force is comparably low. The particles in the deep layer are hardly affected by the erosion of the water flow, but the micro-fluctuation phenomenon occurs with the collision of the particles in the surface layer, as the force is stronger.

## 5. Energy Dissipation Analysis

Thermodynamic principles indicate that energy transformation is an essential factor for the physical change process of matter, and matter destruction is a state instability phenomenon driven by energy [43]. In discrete element theory, the energy dissipation during the erosion failure process of a slope is mainly reflected in the damage and fracturing

of the cohesion model between a gully and nearby soil particles, which reflects the internal mechanism of erosion and erosion evolution. Assuming a closed system without heat exchange with the surroundings, the following equation can be obtained for the slope scour system, according to the first law of thermodynamics:

$$U = U_d + U_e = U_{pre} + \sum_{N_p} \gamma^{(p)} V^{(p)} \Delta x \tag{19}$$

where $U$ is the total input energy generated by the work of the slope body force during the erosion process, $U_d$ is the dissipated energy in the dynamic response process of the slope, $U_e$ is the elastic strain energy stored in the slope particles that can be released, $U_{pre}$ is the upper total input energy accumulated in a calculation time step, $\gamma^{(p)}$ and $\Delta x$ are the particle weight and displacement in the current time step, respectively, $N_p$ is the number of particles, and $V^{(p)}$ is the particle volume. The particle contact model in the PFC model is a parallel bonding model. The elastic strain energy, $U_e$, is calculated from the particle strain energy $U_c$ and the parallel bonding strain energy $U_{pb}$, according to the equations:

$$U_e = U_c + U_{pb} \tag{20}$$

$$U_c = \frac{1}{2} \sum_{N_c} (|F_i^n|^2/k^n + |F_i^s|^2/k^s) \tag{21}$$

$$U_{pb} = \frac{1}{2} \sum_{N_{pb}} [|\overline{F}_i^n|^2/(A\overline{k}^n) + |\overline{F}_i^s|^2/(A\overline{k}^s) + |\overline{M}|^2/(I\overline{k}_n)] \tag{22}$$

where $N_c$ and $N_{pb}$ are the number of contact force chains and the number of parallel bonds, respectively; $i$ denotes the $i$th contact chain; $F_{in}$ and $F_{is}$ are the normal contact force and tangential contact force, respectively; $\overline{F}_i^n$, $\overline{F}_i^s$, and $\overline{M}$ are the normal parallel bonding force, the tangential parallel bonding force, and the parallel bonding moment, respectively; $k_n$ and $k^s$ are the contact normal stiffness and contact tangential stiffness, respectively; $\overline{k}^n$, $\overline{k}^s$, and $\overline{k}_n$ are the stiffness corresponding to each parallel bonding force; and $A$ and $I$ are the cross-sectional area and the moment of inertia of the parallel bond, respectively.

The dissipated energy $U_d$ of the final model can be obtained from Equation (19) by substituting and solving various equations. According to the first law of thermodynamics and the particle flow with an actual energy monitoring system, the instantaneous kinetic energy of the gully can be determined.

*5.1. Kinetic Response*

The kinetic energy response of the gully A–C sections in the model during erosion is shown in Figure 17. The kinetic energy curves of Sections A and C show a stepped course. Integrating the previous analyses, the stepped course of the curves records the erosion damage mode of the No. 3 gully. We can reasonably divide the entire process of erosion damage into three stages, corresponding to the periods of 0–20 min, 20–37 min, and 37–60 min, respectively (Figure 17). The kinetic energy changes at Section A during the first and second stages are small, where each has an energy increase of about 0.5 MJ (Figure 17a). During the third stage, the kinetic energy shows a very large variation, with values ranging from 0.5 MJ to 8.8 MJ. The sharp increase in kinetic energy at this stage was caused by the destruction of soil particles at the bottom and top of the slope, along with the accumulation and collision of many loose particles. Meanwhile, profile C increased to a maximum of 8.9 MJ in the first stage (Figure 17c), related to the initial scour failure in the slope, indicating that Section C is more susceptible to scour damage. The kinetic energy increased continuously, from 5 MJ to 16.2 MJ, in the third stage of erosion damage in Section C, then subsequently decreased to about 7.4 MJ and remained stable. These data indicate that the kinetic energy response at Section C was faster than at Section A, and the energy was higher, with a difference of about 7.4 MJ. Section B was the most important

section for gully failure. The curve clearly documents that the first stage was mainly the failure stage in the center of the slope, where its instantaneous kinetic energy attained a maximum of 25 MJ (Figure 17b). Towards the second and during the third stage, the instantaneous kinetic energy decreased, as the fluid mainly affected Section B, gradually reducing the bonding strength between the particles. Therefore, the instantaneous kinetic energy decreased during the break-up of the particles.

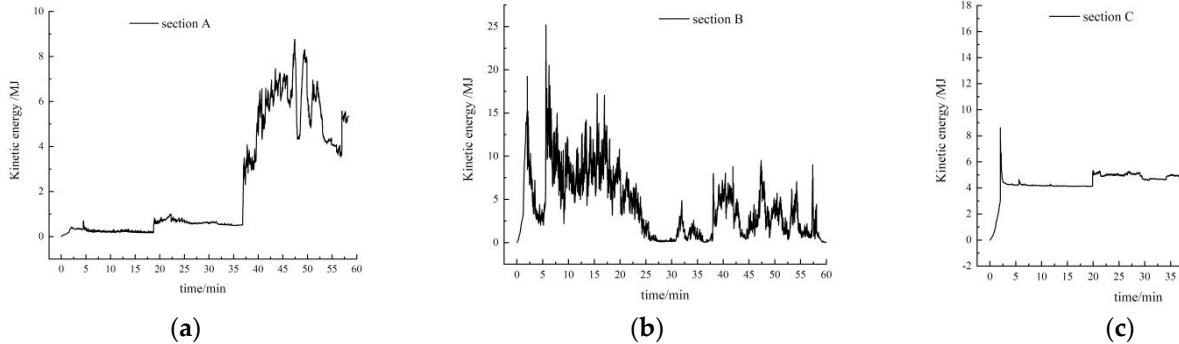

**Figure 17.** Variation in Kinetic Response in the three studied sections. (**a**) Section A (**b**) Section B (**c**) Section C.

In summary, the maximum kinetic energy of the No. 3 gully was concentrated in the erosion damage during the first stage slope, where the maximum kinetic energy reached 25 MJ in Section B. Section C presented a maximum kinetic energy of 16.2 MJ in the later failure of the top and bottom of the slope.

*5.2. Energy Dissipation*

Combined with the equations, the energy dissipation curves of the final three profiles were obtained, as shown in Figure 18. The highest total energy was generated in Section B, and the dissipated energy reached a maximum value of 37 MJ (Figure 18). The maximum dissipated energy in Sections C and Sections A was 28.5 MJ and 13 MJ, respectively. The data reflect three stages of erosion damage:

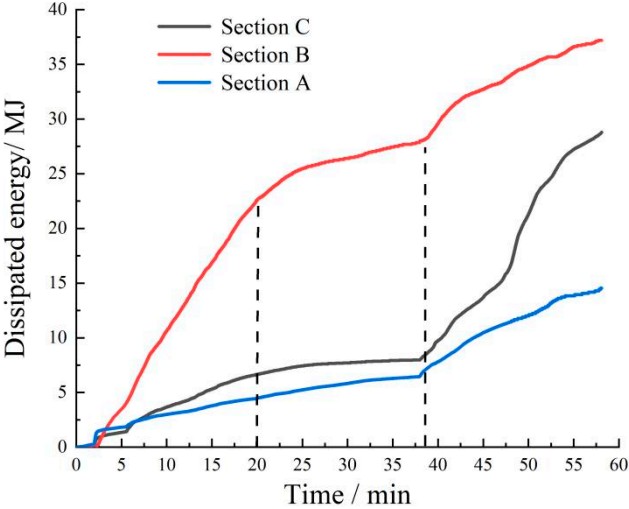

**Figure 18.** Comparison of Energy Dissipation in the three studied sections.

The catchment–fracture stage is the initial one, which was mainly caused by the catchment on both sides of the slope and the top of the slope to the center of the slope. The contact model between the particles in the center of the slope was preferentially fractured. The energy dissipation rate in the valley was comparably high, and the increase in the

dissipated energy reached 22.5 MJ. The dissipated energy on both sides of the gully grew slowly, with a growth amount of 4–7 MJ.

The erosion–accumulation stage represents the second phase. The fluid reduced the strength between the particles during this stage, and the fracture phenomenon in the center of the slope gradually extended to the upper and lower sides, accompanied by the accumulation of particles. The dissipative energy moderately increased, by 2.5–5 MJ, during this stage.

The third phase is the piping–penetration failure stage, which comprises the formation of a complete gully. The piping phenomenon at the top of the slope and the fracture–accumulation–collision at the bottom of the slope led to a sharp increase in the dissipated energy growth rate of Section C. Therefore, the dissipated energy of profile C increased rapidly from 7.6 to 28 MJ (Figure 18). In contrast, the dissipated energy in Sections B and A increased slowly, from 27.7 MJ to 37 MJ in Section B, and from 6 MJ to 14.6 MJ in Section A.

The three stages simultaneously reflect the progressive cumulative energy consumption mechanism in the development of erosion damage. In the first two stages, the dissipated energy curves for Sections C and A were similar. However, due to the formation of lateral tributaries, the dissipated energy of Section C increased sharply in the last stage, with its final value exceeding that of Section A by 15 MJ. The course of the energy dissipation curves is closely related to the failure mode. The extent of the overall energy dissipation is controlled by the position of the slope and the different cross-sections of the gullies, and is additionally affected by the fluid. Moreover, striking differences in the changes in the dissipated energy were recognized in the three stages. Furthermore, the energy dissipation of the No. 3 gully during water flow erosion was concentrated in the catchment–fault stage (the first stage), in the central section of the slope (profile B). Another part of the energy dissipation was concentrated in profile C, regarding the slope bottom accumulation and the slope top piping–penetration failure stage (the third stage).

## 6. Conclusions

Based on the improved erosion shear failure theory, we can calculate the erosion depth at any position of the soil as a function of time. The implemented shear stress of water flow can be applied to PFC simulation, thereby solving the problem associated with the particle velocity error. The erosion depth calculated using the improved equation was in good agreement with the erosion depth simulated by PFC, and both were basically consistent with the actual failure depth of the studied gully (No. 3) in the case study: a Guizhou coal gangue slope in Southwestern China. Therefore, the proposed calculation and modelling methods provide valuable new tools for engineering disaster early warning, as well as a simulation method for extreme precipitation events in geotechnical slopes (spatio-temporal connections, forecasting, generation, impact analysis, and vulnerability and risk assessment).

The most serious erosion damages occurred at the mid-slope position, followed by those at the top of the slope. The accumulation of particles at the bottom of the slope was striking, and the degree of erosion on the slope surface was the lowest. The erosion damage type for the No. 3 gully was intermittent fragmentary damage. The following failure sequence was determined: initial fracture in the center of the slope, accumulation of continuous faults at the bottom of the slope, piping and failure at the top of the slope and, finally, the formation of a large-scale valley-type gully. The entire process of erosion damage could be divided into three stages: catchment–fracture, erosion–accumulation, and piping–penetration failure.

The fracture of the force chain was the most severe in the first stage of erosion, where the coordination number decreased from 5.3 to 4.0, while the porosity increased from 0.42 to 0.45. In the third stage of erosion, the fracture of the force chain between the bottom and the top of the slope was striking, and the value of the coordination number decay reached 1.0–1.2. The peak porosity in the X2 region increased by up to 20%, forming an "M"-shaped course of the curve in the y-direction. Lateral erosion and expansion occurred at the mid-

slope, leading to a tendency to form tributaries, and also serving as the main condition for the formation of multiple ravines. This phenomenon should be further studied in the future.

The unbalanced force value gradually increased from the top to the bottom of the slope. The degree of fluctuation was gradually severe, increasing to 16,000 N. The unbalanced force was generally lower in Section A than in Section C, and the direction of the water flow is more inclined in the x-direction. After ca. 37 min of water flow erosion, the unbalanced force reached 7500 N, inducing lateral erosion and expansion of the gully. The results indicated that, in engineering management, more attention should be paid to the lateral expansion and damage effects caused by rain erosion, in order to effectively avoid further debris flow and landslide disasters.

The maximum kinetic energy of the erosion damage in No. 3 gully was concentrated in the first stage (25 MJ). The energy consumption of No. 3 gully during the water flow erosion was concentrated in the first stage of the central section of the slope (Section B). The other part was focused in the third stage, at the toe and top of the slope (Section C). Accordingly, we suggest adding a section of anti-erosion reinforcement treatment from the central section of the slope to the toe of the slope.

**Author Contributions:** Conceptualization, Y.T.; Formal analysis, F.Z.; Funding acquisition, X.L.; Investigation, J.L., D.X. and Q.F.; Project administration, Q.F.; Software, Y.T. and J.L.; Supervision, X.L.; Validation, Y.T. and D.X.; Writing—original draft, Y.T.; Writing—review & editing, Y.W. All authors have read and agreed to the published version of the manuscript.

**Funding:** This work was financially supported by Research on coordinated control technology of cadmium pollution in farmland soil and groundwater in Chengdu Plain (Project No. 2021YFN0128).

**Data Availability Statement:** The full data set generated for this study is available from the corresponding author on request.

**Conflicts of Interest:** The authors declare that they have no conflict of interest.

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
