# Peer review of "Characterization of the Erosion Damage Mechanism of Coal Gangue Slopes through Rainwater Using a 3D Discrete Element Method: A Case Study of the Guizhou Coal Gangue Slope (Southwestern China)"

_applsci, doi:10.3390/app12178548_

Round 1

Reviewer 1 Report

The topic is interesting. Some suggestions:

- Section 2.1 appears a description of the study area, not a “Research background”

-line 121: “gp” symbol is missing and italics in the sentence “as the soil…” does not seems to make sense. Please fix it

-it would be preferable if conclusion included more qualitative rather than only quantitative  considerations. 

- English proofreading seems to be necessary. E.g., “this study we take” (line 17)

Author Response

We are very pleased to receive your comments. We will revise and explain your comments one by one, as follows:

  • Thank you for pointing out the issue, we have revised it to "Description of the Study Area" at Section 2.1.
  • Missing symbols on line 121 have been added, and the sentence "as the soil" is fixed.
  • Thank you very much for your advice. We have refined certain data and expressions in the Conclusions. (Lines 461, 474, 478, 485) Also, we want to express our own opinion. In the conclusion, we summarize the process of erosion damage as a model (here is a qualitative analysis), and have qualitative analysis in each analysis method. However, we feel that appropriate data descriptions can better reflect the authenticity and scientificity of the conclusions, so we did not delete a large number of quantitative data. Hope you can understand this.
  • We have done an English polish for your journal varsity.

Reviewer 2 Report

This is yet another study in which a discrete-element method is implemented to simulate deformation in soil bodies. The intrinsic flaw is that the characteristics of the PFC elements have nothing to do with those of the soil grains. And in fact, one normally operates an idealization, which is not unlike that made in continuum-based models. Nonetheless, while the latter can well capture stresses and strains, discrete-element models are more apt at capturing macroscopic kinematics, whereas fine-scale deformations and internal forces at specific locations are hardly obtainable straightforwardly. Surely, soil moves by gravity and thus, provided suitable conditions, all PFC simulations will produce reasonable shapes and landslide kinematics. On the other hand, they cannot say anything about internal stresses (including pore water pressures) and strains. The procedure for upscaling/downscaling parameters (that in PFC are completely different to usually used soil parameters) is also unclear, and in this work the authors do not seem to have produced clear upscaling (from lab experiments to model) and downscaling (from model to internal soil parameters) laws, the reliability of which can be tested and discussed. In summary, I do not think this work is a significant contribution to the literature and do not recommend it for publication.

Author Response

Thank you for your important comments on our research content. First of all, we express our immense honor and gratitude. Secondly, we think there are some points of view to share and communicate with you, and hope you will give us a chance to publish. Since we did put a lot of effort and time into this research work, thanks again for your understanding, here are our thoughts and corrections:

  1. The PFC particle flow method regards materials (such as rock, sand, clay, etc.) as discontinuous, and regards the particulate matter composing the material as an independent basic unit, and the interaction between particles reflects the macroscopic nature of the material. mechanical properties. The particle flow numerical analysis method defines a material as a particle aggregate composed of finite particles, a particle is a rigid body with mass, and in three dimensions, a particle is a spherical particle with a unit mass. The basic idea of the discrete element method is to obtain the generalized unbalanced force of the particle through the force-displacement law, and then use Newton's second law to calculate the particle motion behavior under the action of the unbalanced force. The particle flow method uses Newton's second law alternately with the force-displacement law to perform cyclic calculations. Therefore, the deformation and internal force between particles can be obtained through the internal monitoring of the system. In this way, for the simulation of soil slopes, the PFC method is the most suitable (many literatures have also confirmed this, see literature: “10.3389/FMATS.2021.705453”,“10.4028/www.scientific.net/AMM.138-139.459”,“10.1007/s10706-018-0509-8”). Just like you said, the PFC model itself is ideal because of environmental conditions and to improve computational efficiency. But we still strive to get useful data within a reasonable margin of error. In addition, the mechanical properties of the soil slope model established by PFC are determined by the contact model between internal particles, which involves the problem of mesoscopic parameters. Although the internal mechanism of the PFC model has nothing to do with the real soil slope characteristics, the reflected mechanical parameters and mechanical characteristics are the bridges (c, φ, etc.) that establish the two. In the past ten years of research, countless scholars have used the mechanical parameters obtained from the real triaxial test to compare with the results of the PFC triaxial test (see References [24-27]). The purpose of this is to debug the PFC meso-parameters with the same mechanical parameters (which we also mentioned in the text), so that the established model is reasonable and realistic. Although the model is slightly simplified, reasonable data can be obtained on this basis. And we also verified through field data later.

2.At present, many scholars tend to use continuum methods (e.g., FEM, FDM, PFEM) in their research, but these methods present certain drawbacks; for example, they often rely on highly simplified constitutive equations that specify properties. However, there are many parameters in the constitutive equations for accurate prediction, and it is very difficult and cumbersome to calibrate these parameters. In the process of calculation, it is often difficult to calculate convergence when considering plastic problems. On the other hand, the continuum calculation method cannot effectively handle, calculate, and simulate fracturing and large deformation problems.The advantage of PFC is that it can simulate material transport and stress transfer in soil, as well as the deformation, expansion, and extension of rocks and soil in a relatively simple way. It can monitor the position and number of particles generated during the failure of a sample, the stress and strain inside the sample, the shape of the soil when it is deformed, and the strength of the model. The process of dynamic destruction of the macroscopic model can also be observed. Accordingly, this method can realize observations at both the micro- and macro-scale for soil simulation material mechanical and model testing.

  1. In this paper, the gully is not formed by the action of gravity, but the CFD (Computational Fluid Dynamics) fluid module is established to make the model surface destroyed under the action of fluid. Prior to this, we conducted a large number of stability tests on the model to ensure that the unbalanced force between particles was less than 105N before applying fluid action. As for the information of internal stress and strain, this paper does not have a specific analysis, but focuses on the analysis from the perspective of failure mode, porosity change, unbalanced force and energy. The monitoring and analysis of stress and strain will be highlighted in our next research work.
  2. 对于颗粒间尺寸效应,我们在正文中做了一些补充(见第 181 行)。我们丰富了参考文献(参见参考文献 [36-39])。对粒度影响的研究 [33-39] 表明,对于模型粒度与平均粒度之间的比率 > 30,粒度的适当变化不会影响模拟结果。该模型经过反复检查和调试。在提高计算效率和保证雨害模型合理性的基础上,颗粒物平均粒径为1.9m,最小模型尺寸(66m)与(1.9m)之比为35,最大模型尺寸为35。尺寸(130m)与(1.9m)的比值为68。这两个数据都远大于标准值30,因此该模型中的粒径是合理的。

最后,再次感谢您提出宝贵意见。粒子流数值模拟和工程地质分析不是一件容易的事,希望您再给我们一次机会。

请参阅附件。

Reviewer 3 Report

The manuscript is overall interesting. However, some issues are found throughout the paper. Therefore, according to this Reviewer, a moderate revision would be necessary before the paper can be further considered for possible publication in Applied Sciences. All details are summed up in the following.

Required changes:

a)      Despite understandable, English needs some improvements.

b)      Originality/novelty of the study proposed. This issue is very important and should be better clarified and well highlighted in the text.

c)      Introduction is too short, and should be extensively improved. The influence of rainfall infiltration on slope stability mentioned at lines 54-55 should be discussed with more details, including a larger number of pertinent references. For the sake of completeness, the Authors could refer to the following papers:

10.1007/s10346-015-0647-5

10.1061/(ASCE)GT.1943-5606.0002877

10.1016/j.geomorph.2017.03.031

d)       In the current version it is not clear how this manuscript can be useful for practitioners.

e)             Quality of figure is quite poor, in the current version.

Author Response

Thank you very much for your revisions. We will make the following revisions and explanations to your report:

  • We have revised the English writing of this article on the "MDPI Author Services" agency.
  • The innovations of this paper are as follows: 1. The internal mechanism problem of rainfall damage of coal gangue slope is solved by the simulation method of dynamic damage, and the method of energy is introduced to conduct in-depth research on the damage mechanism. 2. This paper optimizes the scouring shear failure theory and combines it with the discrete element PFC3D method, which verifies the rationality and provides a new idea for the PFC flu-id-structure interaction method. 3. This paper enriches the modeling methods of PFC coal gangue slope model, and pro-vides a valuable reference for early warning and simulation of coal gangue slope engineering disasters.(Modified at line 102)
  • Thanks for your comments, we have enriched the introduction (see lines 53-70; lines 75-85; lines 102-107).
  • How this manuscript is useful to practitioners can be obtained from Innovation Points as follows:1. The internal mechanism problem of rainfall damage of coal gangue slope is solved by the simulation method of dynamic damage, and the method of energy is introduced to conduct in-depth research on the damage mechanism. 2. This paper optimizes the scouring shear failure theory and combines it with the discrete element PFC3D method, which verifies the rationality and provides a new idea for the PFC flu-id-structure interaction method. 3. This paper enriches the modeling methods of PFC coal gangue slope model, and pro-vides a valuable reference for early warning and simulation of coal gangue slope engineering disasters.
  • Thank you very much for your comments. We have tried our best to improve the quality of the pictures. However, due to the compression of pictures in word, the current picture clarity is not particularly high. We will continue to improve it later.

Author Response

Thank you for your comments, we will carefully revise and explain your question as follows:

1. For the comparison of the PFC method with other similar methods, our views are as follows (and added in the text with modifications):

Nowadays, many scholars like to use continuum methods (FEM, FDM, PFEM) for their research work, but there are also certain drawbacks. They often rely on highly simplified constitutive equations that specify properties. However, there are many parameters of the constitutive equations for accurate prediction, and it is very difficult and cumbersome to calibrate the parameters. In the process of calculation, it is often difficult to calculate convergence when encountering plastic problems. On the other hand, the continuum calculation method cannot effectively handle, calculate and simulate the fracture and large deformation problems. In addition, in recent years, the SPH (Smoothed Particle Hydrodynamics) method is popular at home and abroad to study the motion of the fluid. This method is a meshless adaptive Lagrangian particle method. It is more complicated, and it is not suitable for the simulation of monitoring points of slope stress and strain. It should be noted that the smooth search function plays an important role in the SPH method because it determines the precision and computational efficiency of the function expression. However, such a function requires a complex algorithm and time to complete, and there are certain drawbacks.

The advantage of PFC is that it can simulate the material transport and stress transfer in the soil, as well as the deformation, expansion and extension of the rock and soil in a relatively simple way. It can monitor the position and number of particles generated during the failure of the sample, the stress and strain inside the sample, the shape of the soil when it is deformed, and the strength of the model. The process of dynamic destruction of the macroscopic model can also be observed. Accordingly, the method can realize the observation on the micro- and macro-scale of soil simulation material mechanical test and model test.

2. The document [20] has been cited in the text, see line 75.

3. We have changed "Prandtland" to "Prandtl",see line 142.

4. We have modified line 121 and it is now on line 154.

5. Thank you, we have modified equation (10).

6. As you said, we changed Equation (10) into a different expression, and expressed it with a differential expression, we can get Equation (12).

7. In equation (19), V(p) is the particle volume.

Round 2

Reviewer 2 Report

I thank the authors for their response but my questions were not answered and I stand with my opinion that the manuscript should not be published. 

Author Response

Thank you for your reply, in response to your previous question, we would like to answer the following:

Question 1:  The intrinsic flaw is that the characteristics of the PFC elements have nothing to do with those of the soil grains.

Response 1:  For the scattered medium such as coal gangue soil, the deformation is not caused by the deformation of the particles themselves, but by the relative movement between the internal particles changing the position of the particles. Therefore, PFC compares soil particles to rigid bodies, which is reasonable for the overall calculation. Furthermore, the contact model between particles and particles is the key to determine the characteristics of simulated soil materials. In this paper, we choose the parallel bonding model as the main contact model. The parallel bonding model can be realized by inter-particle bonding for soil-like bonded materials. The parallel bond model is treated as a pair of springs with constant normal and tangential stiffness, acting parallel to the linear element. When the bond is formed, the relative movement of the particles will generate force and moment at the contact, and when the force or moment is greater than the bond strength, the bond will break. By comparing with the results of the PFC-indoor real triaxial mechanical test of coal gangue, it is found that the parallel bonding model can well reflect the macroscopic mechanical behavior of soil materials. The research object of this paper is coal gangue particles, and the research content is particle breakage, so the mutual contact between particles adopts a linear contact model, and the damage inside the particles adopts a parallel bonding model. When the parallel bonds in the particle clusters are broken, the contact between adjacent particles is converted into linear contact, which facilitates the analysis of meso-mechanical behaviors such as contact force and coordination number.

Question 2:  And in fact, one normally operates an idealization, which is not unlike that made in continuum-based models.

Response 2:  Both discontinuous discrete element and continuum methods belong to numerical simulation methods. The advantage of PFC is that it can simulate the movement and stress transfer laws of particles inside discrete materials, as well as the deformation, expansion and extension processes of rock and soil. This article is based on the above advantages of PFC to simulate the rain erosion failure mechanism of coal gangue slope. Then, the continuum method cannot simulate the fracture and large deformation problems well. At the same time, as a kind of heterogeneous granular material, the continuum method cannot well reflect the interaction between the particles. Therefore, this paper chooses PFC as a research tool to carry out related research.

Question 3:  “Nonetheless, while the latter can well capture stresses and strains, discrete-element models are more apt at capturing macroscopic kinematics, whereas fine-scale deformations and internal forces at specific locations are hardly obtainable straightforwardly. Surely, soil moves by gravity and Thus, provided suitable conditions, all PFC simulations will produce reasonable shapes and landslide kinematics. On the other hand, they cannot say anything about internal stresses (including pore water pressures) and strains.”

Response 3:  As you said, discrete elements are better for simulating the macroscopic kinematics of materials. Although PFC simulates the interaction between soil particles and their motion laws by using Newton's second law and force-displacement law alternately, PFC can obtain the local or global stress and strain of the model through the "measurement region" method. mentioned in the article. 

  Before simulating the failure of soil movement, this paper firstly established a three-axis numerical sample of the soil, and calibrated the meso-parameters of the coal gangue soil material. By comparing the numerical value with the actual stress-strain data, the PFC meso-parameters used in this paper can reasonably simulate the relevant characteristics of coal gangue soil materials. Then, under the meso-parameters used in this paper, a coal gangue slope model is established, and the related research in this paper is carried out. In this paper, the gully is not formed by the action of gravity, but the CFD (Computational Fluid Dynamics) fluid module is established to make the model surface destroyed under the action of fluid. Prior to this, we conducted a large number of stability tests on the model to ensure that the unbalanced force between particles was less than 10-5N before applying fluid action. Then, the coal gangue slope under the action of hydraulic scouring is analyzed from the perspective of failure mode, porosity change, unbalanced force and energy.

Question 4:  “The procedure for upscaling/downscaling parameters (that in PFC are completely different to usually used soil parameters) is also unclear, and in this work the authors do not seem to have produced clear upscaling (from lab experiments to model) and downscaling (from model to internal soil parameters) laws, the reliability of which can be tested and discussed. ”

Response 4:  For the related process of model particle size determination, we have made some additions in the text (see line 179). And we enriched the references (see references [36-39]). In the discrete element simulation research, because the size of the actual soil particles is too small, it is almost difficult to use the actual soil particles for numerical calculation. Therefore, in the PFC simulation, the particle size is often enlarged, and the specific magnification factor It is usually determined based on the needs of the research itself and the computing resources of the hardware. Studies on the effect of particle size [33–39] show that for ratios between model particle size and average particle size > 30, appropriate changes in particle size do not affect the simulation results. The ratio of particle size to average particle size used in this model is 68, which is determined without affecting the simulation accuracy and taking into account the speed of numerical calculation. The particle size of the coal gangue soil triaxial test used in the meso-parameter calibration in this paper is consistent with the particle size of the final numerical model of the coal gangue slope. Therefore, the ratio of the particle size to the average particle size selected in this paper is reliable.

In response to your comments, we have responded carefully and rigorously, and we very much hope that our article will be recognized by you and published.
